# Somatic Development Disorders in Children and Adolescents Affected by Syndromes and Diseases Associated with Neurodysfunction and Hydrocephalus Treated/Untreated Surgically

**DOI:** 10.3390/ijerph19095712

**Published:** 2022-05-07

**Authors:** Lidia Perenc, Agnieszka Guzik, Justyna Podgórska-Bednarz, Mariusz Drużbicki

**Affiliations:** Institute of Health Sciences, University of Rzeszow, 35-959 Rzeszów, Poland; aguzik@ur.edu.pl (A.G.); jpodgorska@ur.edu.pl (J.P.-B.); mdruzbicki@ur.edu.pl (M.D.)

**Keywords:** hydrocephalus, neurodysfunction, developmental disorders

## Abstract

Background: This study was conducted to evaluate the co-occurrence of hydrocephalus treated/untreated surgically and congenital nervous system disorders or neurological syndromes with symptoms visible since childhood, and with somatic development disorders, based on significant data obtained during admission to a neurological rehabilitation unit for children and adolescents. Methods: The study applied a retrospective analysis of data collected during hospitalization of 327 children and adolescents, aged 4–18 years, all presenting congenital disorders of the nervous system and/or neurological syndromes associated with at least one neurodysfunction that existed from early childhood. To allow the identification of individuals with somatic development disorders in the group of children and adolescents with hydrocephalus treated/untreated surgically, the adopted criteria considered the z-score values for body height, body weight, head circumference, body mass index, and head circumference index. Results: Treated/untreated hydrocephalus was observed in the study group at the rates of 8% and 0.9%, respectively. Among 239 patients with cerebral palsy, 9 (3.8%) had surgically treated hydrocephalus, 17 (70.8%) of 24 patients with neural tube defects also had hydrocephalus treated with surgery, and 3 (12.5%) of 24 patients with neural tube defects had untreated hydrocephalus. This medical condition was a more frequent comorbidity in subjects with neural tube defects compared with those with cerebral palsy (*p* < 0.001). Subjects with untreated hydrocephalus most frequently presented macrocephaly (*p* < 0.001), including absolute macrocephaly (*p* = 0.001), and with tall stature (*p* = 0.007). Excessive body mass co-occurred more frequently with surgically untreated hydrocephalus, but the relationship was not statistically significant (*p* = 0.098). Conclusions: Surgically treated hydrocephalus occurred in patients with cerebral palsy and neural tube defects, and untreated hydrocephalus was present only in patients with neural tube defects. Untreated hydrocephalus negatively changed the course of individual development in the studied group of children, in contrast to surgically treated hydrocephalus.

## 1. Introduction

Various neurodysfunctions can result from nervous system disorders; encephalopathy may, but does not have to occur, in these cases. Numerous factors of different origin can lead to encephalopathy, which is how brain damage is generally described. Since it contributes to behavioral and cognitive dysfunctions, which are frequently associated with motor impairments, encephalopathy makes it difficult to start school education [1,2].

Hydrocephalus is associated with increased cerebrospinal fluid pressure and enlargement of the ventricular system of the brain [3,4,5]. Only in infancy, before fontanelle ossification and closure of the sutures, the symptoms of intracranial hypertension associated with hydrocephalus are accompanied by a progressive macrocephaly [4,5,6]. Hydrocephalus that occurs in childhood is not a homogeneous disease. It can start as a complication of various conditions in the prenatal and postnatal period [4,5]. It can present with varying degrees of severity. Neurosurgical procedures include the insertion of a drainage system or endoscopic ventriculostomy [5,6]. Some children with mild hydrocephalus require only observation. The long-term effects of the neurosurgical treatment of hydrocephalus are varied [5,6].

The most common disorders of somatic development are related to abnormal head size, impaired growth, and abnormal weight–height ratio. To assess these disorders of somatic development, it is necessary to compare the measurements collected from the subject to a relevant biological reference system [7]. Microcephaly and macrocephaly are skull development disorders (head size) and, at the same time, they are treated as clinical symptoms [8,9]. These two developmental disorders can be classified as relative or absolute. In addition to the relative microcephaly/macrocephaly, which corresponds to a proportional decrease/increase in head circumference and body height, there isabsolute microcephaly/macrocephaly, reflected by an isolated decrease/increase in head circumference [8,10]. As an example, growth disorders may be reflected by two opposing conditions: short stature and tall stature [11,12,13]. Based on the assessment of the varied weight–height ratios, it is possible to evaluate the nutritional status [14,15,16]. In clinical conditions and scientific research, different definitions of somatic development disorders and various biological reference systems, or reference values, are intentionally used [7,8,9,10,11,12,13,14,15,16,17,18,19]. The present study was designed as a continuation of previously reported research focusing on somatic development disorders in a group of children and adolescents with neurodysfunction [9,10,11,15,16]. The prevalence of the above-mentioned somatic development disorders has already been determined in the entire study group [9,10,11,15,16]. However, the coexistence of hydrocephalus treated/untreated surgically and disorders of somatic development in the study group was not taken into account.

It should be emphasized that these visible developmental disorders can contribute to stigmatization of individuals affected by diseases and syndromes associated with neurodysfunctions [20]. Implementing effective diagnostic and therapeutic measures to reduce the severity of visible developmental disorders may effectively reduce this problem [21].

The aim of the study was to determine the relationship between surgically treated and untreated hydrocephalus and diseases/syndromes related to neurodysfunction. The relationship between somatic development disorders and hydrocephalus treated and untreated was investigated in a group of children and adolescents with diseases/syndromes associated with neurodysfunction, based on significant data obtained during admission to a neurological rehabilitation unit for children and adolescents. It was hypothesized that untreated hydrocephalus affects the course of individual development in the examined group of children and adolescents, in contrast to surgically treated hydrocephalus.

## 2. Materials and Methods

The methodological assumptions presented below are consistent with those presented earlier [9,10,11,15,16].

### 2.1. Participants

This retrospective study is based on information relating to 327 children and adolescents admitted to the Neurological Rehabilitation Ward for Children and Adolescents in Regional Hospital No. 2 in Rzeszow, Poland, in the period 2012–2016, and staying at the Regional Clinical Rehabilitation and Education (KRORE). All patients included in the study were treated during the above period due to congenital nervous system disorders or neurological syndromes, with one or more visible neurodysfunctions since infancy [9,10,11,15,16]. Various additional qualifying criteria were applied, namely, age in the range of 4–18 years (the lower threshold value was selected for the following reasons: (a) a relatively high percentage of children under 4 years of age do not have an established diagnosis [22,23,24], (b) the value is consistent with the methodological assumptions adopted throughout the research project, (c) it matches the age range in the applicable biological frame of reference containing statistical characteristics of body height, body weight, head circumference, body mass index, and head circumference index [25,26]); informed consent obtained from the examined children and adolescents, in addition to their parents/legal guardians; and medical records containing the measurement results related to body height and weight, head circumference and gross motor function classification system (GMFCS), in addition to complete diagnostic data, all of which were acquired during a single admission procedure [9,10,11,15,16].

In principle, patients admitted for rehabilitation in stationary conditions may not have symptoms of intracranial hypertension, so they would not undergo the admission procedure. However, the patients had the described diagnoses: condition after surgery due to hydrocephalus or normotensive hydrocephalus. These diagnoses, similar to the others, weremade by specialist physicians before admission to the ward. Such an approach (recording diagnoses) is necessary to define the rehabilitation program, determine contraindications to rehabilitation, and to immediately react in the event of symptoms of intracranial hypertension (neurosurgical consultation, diagnostics, qualification for surgery, and discharge from the department to the indicated unit). Therefore, we were able to analyze whether treated/untreated hydrocephalus differentiates the development of neurosurgical children.

The following exclusion criteria were used: (a) no diagnosis related to a congenital disorder of the nervous system or a neurological syndrome linked with one or more neurodysfunctions visible from infancy, (b) existing combinations of congenital disorders of the nervous system or neurological syndromes (e.g., Down syndrome, neural tube defect, or phenylketonuria co-occurring with cerebral palsy). Furthermore, patients were excluded if they were not hospitalized within the relevant time frame, or if their records contained more than one admission procedure, or if their age was outside the range of 4–18 years (owing to the lack of biological reference frame that contained statistical characteristics of body height and weight, head circumference, body mass index, head circumference index), or if their hospital records did not contain complete information related to the diagnosis and anthropometric measurements (body weight and height, head circumference) and GMFCS, and if children and their parents/legal guardians did not provide informed consent to participate in the study [9,10,11,15,16].

In the investigated period of 2012–2016, a total of 2637 patients were hospitalized in the Neurological Rehabilitation Ward for Children and Adolescents at the Regional Hospital No. 2 in Rzeszow, Poland, and staying at KRORE. Of these, only 327 patients met all the inclusion criteria. In this group, there were 184 boys (56.3%) and 143 girls (43.7%), with a mean age of 9.7 ± 4.3 years (median 9.0 years; the youngest child was 4 and the oldest was 18 years of age). The study design was evaluated and accepted by the Bioethics Committee of the University of Rzeszow, Poland. The research procedure was carried out according to the relevant guidelines and regulations. Before the start of the study, informed consent was obtained from both the children and their parents or legal guardians, and additionally from the director of the hospital.

### 2.2. Procedures and Data Analyses

The retrospective analysis performed was based on essential patient data such as age, sex, diagnosis, body weight (w), body height (h), and head circumference (hc). All the information was retrieved from the patient records collected at admission [9,10,11,15,16].

Accurate diagnoses were established by specialists such as neurologists, geneticists, endocrinologists, and others, prior to admission to KRORE. On the basis of medical records, it was observed that the young patients presented a variety of disorders and syndromes associated with damage to the nervous system. In other words, all of these were congenital anomalies and/or disorders, with or without encephalopathy, and accompanied by motor defects (neurodysfunctions) visible from early childhood. When dividing patients into subgroups, the criteria defined in the specialist literature (that is, suspected encephalopathy or no encephalopathy, its etiopathogenesis, and nature) were applied [27]. Seven subgroups were distinguished in the entire group, six of them comprised patients with diseases/syndromes usually involving encephalopathy: progressive in metabolic disease (2.1%), progressive epileptic–genetically conditioned epileptic syndromes (0.3%), non-progressive in neural tube defects (7.3%), non-progressive in genetically conditioned diseases (chromosomal aberrations, monogenic diseases, except for neuromuscular diseases) (7.0%), non-progressive toxic (0.3%), and non-progressive in cerebral palsy (73.1%). Finally, a subgroup included patients with diseases generally without encephalopathy, i.e., neuromuscular diseases (9.8%) [9,10,11,15,16,27]. Ultimately, three larger groups were formed based on the smaller groups, i.e., comprising patients with progressive encephalopathy (2.4%), non-progressive encephalopathy (87.8%), and those with neuromuscular diseases (9.8%).

The resulting structure of the study group is presented again. The seven subgroups are characterized from the most frequent to the least frequent. Cerebral palsy was the most numerous subgroup (N = 239, 73.1%) [9,10,11,15,16]. The following syndromes occurred in the subgroup of neuromuscular diseases: hereditary motor and sensory polyneuropathy (N = 8, 2.4%), limb–girdle muscular dystrophy (N = 7, 2.1%), Becker muscular dystrophy (N = 3, 0.9%), Duchenne muscular dystrophy (N = 7, 2.1%), Thomsen disease (N = 1, 0.3%), arthrogryposis multiplex congenita with neuropathy (N = 3, 0.9%), congenital myopathy (N = 1, 0.3%) and spinal muscular atrophy (N = 2, 0.6%) [9,10,11,15,16].

Furthermore, to accommodate for the large variety of neural tube defects [9,10,11,15,16,28,29] and the importance of further surgical treatment [6,9,10,11,15,16,17], the relevant group was divided into subgroups comprising patients operated on myelomeningocele and hydrocephalus, operated only on myelomeningocele, and other cases in which no surgical treatment was administered. The following disorders occurred in the neural tube defect subgroup: patients operated for myelomeningocele and hydrocephalus (N = 17, 5.2%), patients operated only for myelomeningocele (N = 3, 0.9%), state after surgery due to parieto-occipital meningocele (N = 1, 0.3%), Arnold–Chiari malformation (N = 2, 0.6%) and isolated hydrocephalus (N = 1, 0.3%) [9,10,11,15,16]. Both chromosomal aberrations and genetic mutations were present in the subgroup with genetically conditioned disorders. It should be mentioned that some authors point to chromosomal disorders and genetic mutations as factors that cause short stature. Down syndrome and Prader–Willi syndrome also belong to this group, together with other genetically conditioned diseases, including mutations of a single gene [9,10,11,12,13,15,16]. The subgroup that included genetic disorders was the most diverse: Down syndrome (N = 11, 3.4%), Edwards syndrome (N = 1, 0.3%), Phelan–McDermid syndrome (N = 2, 0.3%), Mowat–Wilson syndrome (N = 1, 0.3%), Angelman syndrome (N = 1, 0.3%), DiGeorge syndrome (N = 1, 0.3%), 46,XY,del(X) (q24) (N = 1, 0.3%), Cornelia de Lange syndrome (N = 1, 0.3%), Shwachman–Diamond syndrome (N = 1, 0.3%), Prader–Willi syndrome (N = 1, 0.3%), 46 XX, add(2)(q25) (N = 1, 0.3%), 46XX, del (12) (q24.21q24.23) (N = 1, 0.3%). Progressive metabolic disorders were represented by 2 people (0.6%) with neurodegeneration due to iron accumulation in the brain, mitochondrial protein-associated neurodegeneration, and 1 person (0.3%) each representing the following diagnoses: Pompe disease, long-chain 3-hydroxyacyl-coenzyme A dehydrogenase deficiency, Smith–Lemli–Opitz syndrome, type 1 glucose transporter deficiency, non-ketotichyperglycinemia. There was also a person with Dravet syndrome (0.3%) in the subgroup of progressive genetically determined epileptic syndromes. Finally, foetal alcohol syndrome (N = 1, 0.3%) represented the subgroup of non-progressive toxic encephalopathy [9,10,11,15,16].

Anthropometric indices were calculated based on the results of measurements performed by hospital personnel in accordance with the guidelines applicable in KRORE. Such indices, as body mass index (BMI), in addition to head circumference index (HCI), i.e., a quotient of head circumference and body height (hc/h), were calculated for each patient individually. To allow assessment of the developmental deficits that occurred in the examined children, based on the parameters mentioned above, z-scores were calculated for body height (z-score h), body weight (z-score w), head circumference (z-score hc), body mass index (z-score BMI), and head circumference index (z-score HCI). Previously published normative values were used as a reference system [25,26]. In addition, z-score indices were used to identify the developmental disorders described below. The previously published articles have already presented the detailed statistical characteristics of z-score indices. The arithmetic mean and standard deviation have been recalled: z-score h = −1.23 ± 1.98, z-score w = −0.78 ± 1.98, z-score hc = −0.53 ± 2.14, z-score BMI = −0.33 ± 1.76, z-score HCI = 0.90 ± 2.04 [9,10,11,15,16].

Based on two evaluation criteria, that is, those used in dysmorphology and those traditionally applied in clinical practice, children and adolescents with normal head size, microcephaly, and macrocephaly were identified in the study group [8,9]:Head size—dysmorphology classification (hc): normal head size (−3 z score hc ≤ 3) and abnormal development of the cranium: microcephaly (z score hc < −3) and macrocephaly (z-score hc> 3),Head size—traditional classification (hc): normal head size (−2 ≥ z-score hc ≤ 2) and abnormal cranial development: microcephaly (z-score hc < −2) and macrocephaly (z-score hc > 2).

Furthermore, the authors proposed a modification, i.e., in addition to head circumference (hc), they applied the ratio of head circumference and body height (HCI). This means that differences in body proportions corresponding to head circumference (hc) and body height (h) were taken into account. To make an assessment based on all the above criteria, the values calculated for each subject included a head circumference z-score (z-score hc), and a head z-score ratio of head circumference and body height (z-score HCI). The criteria for identifying an abnormal head size [10]:The abnormal head size—dysmorphology classification based on head circumference and head circumference index (hc&HCI): relative microcephaly (z-score hc < −3 and z-score HCI ≠ (<−3), absolute microcephaly (z-score hc < −3 and z-score HCI <−3), relative macrocephaly (z-score hc> 3 and z-score HCI ≠ (> 3)), absolute macrocephaly (z-score hc> 3 and z-score HCI > 3);The abnormal head size—traditional classification (hc&HCI): relative microcephaly (z-score hc < −2 and z-score HCI ≠ (<−2)), absolute microcephaly (z-score hc < −2 and z-score HCI <−2), relative macrocephaly (z-score hc> 2 and z-score HCI ≠ (> 2)), absolute macrocephaly (z-score hc > 2 and z-score HCI > 2).

To identify individuals with short stature, normal body height, and tall stature in the studied group, two evaluation criteria were taken into account. One was based on the approach traditionally used in clinical practice, and the other was in line with the recommendations of the Food and Drug Administration (FDA) [11,18] regarding the recognition of short stature. The criteria of growth disorders:Height—traditional classification (h): normal body height (-2 ≥ z-score h ≤ 2), short stature (z-score h <−2), tall stature (z-score h > 2);Height—proposed classification (h): normal body height (−2.25 ≥ z-score h ≤ 2.25), short stature (z-score h <−2.25), tall stature (z-score h > 2.25).

To assess nutritional status, we applied the BMI z-score and the criteria previously applied in earlier studies [14,15,16]. To correctly differentiate the body proportions, the range −1 ≥ z-score BMI ≤ 1 was adopted. Children with such BMI values have normal weight-to-height ratio, which generally reflects a lack of nutritional status disorders [14,15,16]. A diagnosis of a nutritional disorder, namely, malnutrition [30,31,32], overweight [33,34], or obesity [33,34], is associated with the requirement to implement therapeutic interventions [16,31,32,33,35,36,37]. Classification of nutritional status based on the BMI z-score [14,15,16]:Three categories of nutritional status (BMI): normal state of nutrition (−1 ≥ z-score BMI ≤ 1), body mass deficiency in relation to height (z-score BMI <−1), excess body weight in relation to height (z-score BMI > 1);Five categories of nutritional status (BMI): normal state of nutrition (−1 ≥ z-score BMI ≤ 1), malnutrition (z-score BMI <−1.64), underweight (−1.64 ≥ z-score h <−1), overweight (1 > z-score BMI ≤ 1.64), obesity (z-score BMI > 1.64).

The evaluation of the severity of gross motor development disorders in the study group was based on the gross motor function classification system—GMFCS I-V (I—1 point, V—5 points). Levels I and II of GMFCS were combined as group A, level III corresponded to group B, and levels IV and V were combined as group C (A—1 point, C—3 points). In Poland, each patient with neurodysfunction admitted to a neurological rehabilitation unit for children and adolescents is assessed on the GMFCS scale [9,10,16,38,39,40]. The arithmetic mean and standard deviation are recalled: GMFCS I-V = 2.48 ± 1.30, GMFCS A-C = 1.60 ± 0.86.

At the next stage, the result of the cross-analysis is presented:Co-occurrence of hydrocephalus treated surgically with diseases and syndromes associated with neurodysfunction (Table 1A);Occurrence of hydrocephalus treated surgically in separate subgroups (Table 1B,C);Co-occurrence of hydrocephalus untreated surgically with diseases and syndromes associated with neurodysfunction (Table 2A);Occurrence of hydrocephalus untreated surgically in separate subgroups (Table 2B,C);Co-occurrence of hydrocephalus treated surgically with developmental disorders (Table 3A–H);Co-occurrence of hydrocephalus untreated surgically with developmental disorders (Table 4A–H).

Subsequently, it was assessed whether there was a difference in the level of motor development determined using the GMFCS scale:Between the subgroup with hydrocephalus treated or untreated surgically, and the rest of the examined group (lack of hydrocephalus) (Table 5A,C,E,G);Despite the low numbers, between the subgroups with hydrocephalus treated and untreated surgically (Table 5B,D,F,H).

Dependency analysis was shown as a summary of the number (N) and percentage structure (%) of responses to selected research questions on somatic development disorders in the assessed groups. Adjusted standardised residuals (ASR) are shown next to percentages in the crosstabs. Values greater than 1.96 corresponded to a higher number, and those less than −1.96 represented a smaller number than a random distribution. Statistical inference methods were applied to determine whether the differences wererandom or if they reflected certain regularities in the relevant population. The Chi-square independent test was applied in further analyses due to the nominal nature of the characteristics investigated. The nominal logistic regression model was applied to determine the associations between the dependent qualitative and independent quantitative variables. It was assumed *p* < 0.05 reflected statistical significance. Pearson’s contingency coefficient C (Cp) could take positive values (Cp ≥ 0) only. Some relationships were reflected by Cp distant from 0, while values approaching 1 reflected a perfect association. The Mann–Whitney test was applied to compare ordinal variables between two groups [9,10,11,15,16].

## 3. Results

### 3.1. Basic Percentage Analysis

The frequency of hydrocephalus treated surgicallywas 8.0% (N = 26) and non-treated hydrocephalus was 0.9% (N = 3). Of 239 patients with cerebral palsy, 9 (3.8%) had an additional diagnosis of hydrocephalus treated surgically (Table 1A,B). Among 24 patients with neural tube defects, 17 (70.8%) also had the diagnosis of hydrocephalus treated surgically (Table 1B), and 3 (12.5%) of 24 patients with neural tube defects had the diagnosis of normotensive hydrocephalus not requiring surgery (hydrocephalus untreated surgically) (Table 2B). In the neural tube defects subgroup, the total number of patients with hydrocephalus treated and untreated surgically was 83.3% (20 patients out of 24). Among patients with an additional diagnosis of hydrocephalus treated surgically (N = 26) as primary diagnosis, 34.6% had cerebral palsy (N = 9) and 65.4% had neural tube defects (patients operated for both myelomeningocele and hydrocephalus) (N = 17) (Table 1A,B). Hydrocephalus untreated surgically (N = 3) occurred only in the subgroup of patients with neural tube defects (patients operated only for myelomeningocele (N = 2, 66.7%) and isolated hydrocephalus (N = 1, 33.3%) (Table 2A,B).

### 3.2. Co-Occurrence of Hydrocephalus Treated Surgically withDiseases and Syndromes Associated with Neurodysfunction and Occurrence of Treated Hydrocephalus in the Separate Subgroups

The presence of more/less frequent co-occurrence and statistically significant relationships was found between:Units and syndromes with neurodysfunction and hydrocephalus treated surgically (*p* < 0.001, Cp = 0.624). In the study group, treated hydrocephalus coexisted more frequently with the state after surgery myelomeningocele and hydrocephalus (65.4%, ASR = 14.4) and less frequently with cerebral palsy (34.6%, ASR= −4.6) (Table 1A);Classification with respect to the etiopathogenesis, presence and character of encephalopathy and hydrocephalus treated surgically (*p* < 0.001, Cp = 0.549). In the study group, treated hydrocephalus coexisted more frequently with neural tube defects (65.4%, ASR = 11.8) and less frequently with cerebral palsy (34.6%, ASR= −4.6) (Table 1B).

The presence of more/less frequent co-occurrence and the lack of statistically significant relationships was found between:Classification with respect to the presence and character of surgically treated encephalopathy and hydrocephalus (*p* = 0.140). In the study group, surgically treated hydrocephalus co- occurred more frequently with non-progressive encephalopathy (100%, ASR = 2.0) (Table 1C).

### 3.3. Co-Occurrence of Hydrocephalus Untreated Surgically with Diseases and Syndromes Associated with Neurodysfunction and Occurrence of Untreated Hydrocephalus in Separate Subgroups

The presence of more/less frequent co-occurrence and the presence of statistically significant relationships was found between:Units and syndromes with neurodysfunction and hydrocephalus untreated surgically (*p* < 0.001, Cp = 0.661). In the study group, surgically untreated hydrocephalus coexisted more frequently with the state after surgery myelomeningocele (66.7%, ASR = 12.0) and isolated hydrocephalus (33.3%, ASR = 10.4), and less frequently with cerebral palsy (0.0%, ASR= −2.9) (Table 2A);Classification with respect to the etiopathogenesis, presence and character of encephalopathy and hydrocephalus untreated surgically (*p* < 0.001, Cp = 0.324). In the study group, surgically untreated hydrocephalus coexisted more frequently with neural tube defects (100.0%, ASR = 6.2) and less frequently with cerebral palsy (0.0%, ASR= −2.9) (Table 2B).

No more/less frequent co-occurrence was found and no statistically significant relationships were found between:

Classification with respect to the presence and character of encephalopathy and hydrocephalus not treated surgically (Table 2C).

### 3.4. Co-Occurrence of Hydrocephalus Treated Surgically with Developmental Disorders

The presence of more/less frequent co-occurrence and the lack of statistically significant relationships was found between:Head size—classification of dysmorphology (hc) and hydrocephalus treated surgically (*p* = 0.121). Hydrocephalus treated surgically often co-occurred with macrocephalus (11.5%, ASR = 2.1). Macrocephalus and the absence of surgically treated hydrocephalus co-occurred rarely (3.3%, ASR= −2.1) (Table 3A);Head size—traditional classification (hc&HCI) and hydrocephalus treated surgically (*p* = 0.052). This relationship was close to statistical significance. Absolute macrocephaly and surgically treated hydrocephalus co-occurred frequently (50.0%, ASR = 2.7). Absolute macrocephaly and the absence of surgically treated hydrocephalus co-occurred rarely (13.7%, ASR= −2.7) (Table 3B);The abnormal head size—dysmorphology classification (hc&HCI) and hydrocephalus treated surgically (*p* = 0.051). The relationship was close to statistical significance. Absolute macrocephaly and surgically treated hydrocephalus co-occurred frequently (50.0%, ASR = 2.6). Absolute macrocephaly and the absence of surgically treated hydrocephalus co-occurred rarely (10.4%, ASR= −2.6) (Table 3C);Height—proposed classification (h) and hydrocephalus treated surgically (*p* = 0.084). Correct body height co-occurred less frequently with surgically treated hydrocephalus (53.8%, ASR= −2.2). Short stature co-occurred less frequently with surgically treated hydrocephalus (46.2%, ASR = 2.2) (Table 3E);Height—traditional classification (h) and hydrocephalus treated surgically (*p* = 0.053). The relationship was close to statistical significance. Correct body height rarely co-occurred with surgically treated hydrocephalus (46.2%, ASR= −2.4). Short stature co-occurred more frequently with surgically treated hydrocephalus (50.0%, ASR = 2.4) (Table 3F);Three categories of nutritional status and hydrocephalus treated surgically (*p* = 0.131). Excessive body mass co-occurred more frequently with surgically treated hydrocephalus (34.6%, ASR = 2.2) (Table 3G);Five categories of nutritional status and hydrocephalus treated surgically (*p* = 0.131). Overweight co-occurred more frequently with surgically treated hydrocephalus (34.6%, ASR = 2.1) (Table 3H).

No more/less frequent co-occurrence was found and no statistically significant relationships were found between:Head size—traditional classification (hc) and hydrocephalus treated surgically (Table 3B).

### 3.5. Co-Occurrence of Hydrocephalus Untreated Surgically with Developmental Disorders

The presence of more/less frequent co-occurrence and statistically significant relationships was found between:Head size—dysmorphology classification (hc) and hydrocephalus untreated surgically (*p* = 0.030, Cp = 0.145). Macrocephalus and surgically untreated hydrocephalus co-occurred frequently—(33.3%, ASR = 2.6). Macrocephalus and the absence of surgically untreated hydrocephalus co-occurred rarely (3.7%, ASR= −2.6) (Table 4A);Head size—traditional classification (hc) and hydrocephalus untreated surgically (*p* < 0.001, Cp = 0.285). Macrocephaly and surgically untreated hydrocephalus co-occurred frequently (100.0%, ASR = 5.4). Macrocephaly and the absence of surgically untreated hydrocephalus co-occurred rarely (8.6%, ASR= −5.4). Correct head circumference and surgically untreated hydrocephalus co-occurred rarely (0.0%, ASR= −2.6). Correct head circumference and the absence of surgically untreated hydrocephalus co-occurred frequently (69.1%, ASR = 2.6) (Table 4B). Pearson’s C contingency coefficient (0.285) was higher for the traditionally used definition than for those used in dysmorphology (Cp = 0.145). The definition traditionally used in clinical practice, in this case, better differentiates the relationship between macrocephaly and surgically untreated hydrocephalus (Table 4A,B);The abnormal head size—traditional classification (hc&HCI) and hydrocephalus untreated surgically (*p* = 0.001, Cp = 0.363). Absolute macrocephaly and untreated hydrocephalus co-occurred frequently (100.0%, ASR = 4.0). Absolute macrocephaly and the absence of surgically untreated hydrocephalus co-occurred rarely (14.0%, ASR = −4.0). Relative microcephaly and surgically untreated hydrocephalus co-occurred rarely (0.0%, ASR= −2.1). Relative microcephaly and the absence of surgically untreated hydrocephalus co-occurred frequently (60.0%, ASR = 2.1) (Table 4D);Height—traditional classification (h) and hydrocephalus untreated surgically (*p* = 0.007, Cp = 0.72). Tall stature was more common with untreated hydrocephalus in the study group of children (33.3%, ASR = 3.1) (Table 4F).

The presence of more/less frequent co-occurrence and the lack of statistically significant relationships was found between:The abnormal head size—dysmorphology classification (hc&HCI) and hydrocephalus untreated surgically (*p* = 0.119). Macrocephaly and surgically untreated hydrocephalus co-occurred frequently (100%, ASR = 2.4). Macrocephaly and the absence of surgically untreated hydrocephalus co-occurred rarely (13.2%, ASR= −2.4) (Table 4A);Three categories of nutritional status and hydrocephalus not surgically treated (*p* = 0.098). In the studied group of children, excessive body mass co-occurred more frequently with surgically untreated hydrocephalus (66.7%, ASR = 2.1) (Table 4G).

No more/less frequent co-occurrence was found and no statistically significant relationships were found between:Height—proposed classification (h) and hydrocephalus untreated surgically (Table 4E);Five categories of nutritional status and hydrocephalus not surgically treated (Table 4H).

### 3.6. Gross Motor Function Classification System and Hydrocephalus Treated and Untreated Surgically

There was no statistically significant difference for GMFCS levels (Table 5A–D), but there was for the GMFCS score (Table 5E,H). The lack of significance in the analogous analysis, but with the GMFCS not treated as a number of points but as a level, resulted from the fact that when GMFCS was not considered as scores, then the numbers of patients in particular GMFCS levels became important. In the hydrocephalic group, they were small. GMFCS I-V scores (Table 5E) and GMFCS A–C scores (Table 5G) were significantly higher in the hydrocephalus group treated and untreated, compared with the rest of the study group (*p* = 0.027 and *p* = 0.036, respectively).

## 4. Discussion

The rates of hydrocephalus treated/untreated surgically in the study group: children with units or disease syndromes with neurodysfunction amounted to 8% and 0.9%, respectively (the total incidence of hydrocephalus was 8.9/100). Other data related to the incidence of hydrocephalus can also be found in the literature. The incidence of congenital hydrocephalus among new-borns is 0.4–0.6/1000 [3]. The incidence of hydrocephalus among infants is 1.1/1000 [4]. The annual incidence of idiopathic normal pressure hydrocephalus (Hakim–Adams syndrome) ranges from 0.5/100,000 to 5.5/100,000 [3].

Hydrocephalus can occur with an increase in intracranial pressure and an extension of the ventricular system of the brain. Closed hydrocephalus is associated with birth defects, can complicate myelomeningocele and encephalitis, intracerebral haemorrhage, or the proliferative process. Communicating hydrocephalus is the result of cerebrospinal fluid overproduction, absorption disorders, or venous drainage failure. In the case of normotensive hydrocephalus or atrophy, we do not observe an increase in intracranial pressure, but only an extension of the ventricular system of the brain [4,5]. If hydrocephalus progresses, surgery should be indicated [5,6].

Based on the conducted research, surgically treated hydrocephalus occurred only in the group of non-progressive encephalopathies: it coexisted with neural tube defects, specifically in children with a state after myelomeningocele and hydrocephalus surgery (which in this case was related to methodological assumptions (65.4%) and in children with cerebral palsy (34.6%)). Surgically untreated hydrocephalus also occurred only in the group of non-progressive encephalopathies, but only co-occurred with neural tube defects: state after surgery of myelomeningocele (66.7%) and isolated hydrocephalus (33.3%). The incidence of cerebral palsy is approximately 3/1 000 births [41]. Intraventricular hemorrhage is a frequent complication in extremely preterm and very preterm births. Among all preterm newborns admitted to the Department of Neonatology at Rouen University Hospital, France, in 2000–2013, 122 had intraventricular hemorrhage and were included in the study. High intraventricular hemorrhage grade, low gestational age at birth, and increased head circumference were risk factors for post-hemorrhagic hydrocephalus. The rate of cerebral palsy was 55.9% in the 77 surviving patients. The impact of increased head circumference highlights the need for early diagnosis and treatment [42]. Myelomeningocele is one of the most common congenital abnormalities of the central nervous system (occurring in approximately 1 per 1000 births worldwide) [29,43]. The vast majority of patients with neural tube defects have hydrocephalus [4]. Individuals who survive to birth have their lesions surgically closed, with subsequent management of associated defects, for example, hydrocephalus that often requires shunting [29]. Normal pressure hydrocephalus has long been known to occur in patients with myelomeningocele [44]. Some children with mild hydrocephalus are thought to require only observation [5,6].

Absolute macrocephaly (50.0% regardless of accepted criteria) or short stature (46.2% or 50.0% depending on adopted criteria) often coexisted with surgically treated hydrocephalus in the examined group, and the correlation approached statistical significance. Overweight often occurred (34.6%) with hydrocephalus treated surgically in the study group; however, the correlation wasnot statistically significant. Macrocephaly, namely, absolute macrocephaly (100.0% regardless of accepted criteria) and height (33.3% for traditional criteria), often coexisted with untreated surgically hydrocephalus, and the relationship wasstatistically significant. Excessive body mass often (66.7%) occurred with untreated hydrocephalus in the study group, however, the correlation wasnot statistically significant. Other results of the research conducted among the same group of patients were presented earlier. The study showed that among children and adolescents with neurodysfunction, there were statistically significant relationships between short stature and state after myelomeningocele and hydrocephalus surgery among patients with neural tube defects, short stature, and spastic cerebral palsy among patients with cerebral palsy. Furthermore, the short stature in the study group was associated with hypothyroidism. No statistically significant relationships other than the one presented were found between the coexistence of high stature and the main diagnoses or subgroups discussed in this article [11]. Other studies have shown that the short stature after surgery in the myelomeningocele and hydrocephalus group of patients was attributable to smaller lower extremities [17,45], spinal deformities and scoliosis [45]. Furthermore, excessive fat accumulation occurred in patients with state after myelomeningocele and hydrocephalus and state after surgery of myelomeningocele and hydrocephalus surgery [17,46]. Malnutrition occurred with spastic cerebral palsy among children and adolescents with cerebral palsy [14]. In some cases, such as concomitant hydrocephalus, high growth, kyphoscoliosis, and excessive mobility in the joints, congenital connective tissue disease was also observed. Hydrocephalus in connective tissue diseases is progressive in nature [47]. The incidence of short and tall stature in the general population is similar. The reasons for high growth are: familial tall stature, constitutional advance of growth, aromatase deficiency, estrogen receptor α deficiency, congenital adrenal hyperplasia, growth hormone excess or hyperthyroidism, McCune–Albright syndrome, Beckwith-Wiedemann syndrome, Klinefelter syndrome, Triple X, fragile X, homocystinuria, Sotos and Weaver syndrome [48]. As examples of the cause of macrocephaly, mild family macrocephaly, mitochondrial encephalopathy, 22q11.2 microdeletion [49], andatypical Rett syndrome [50] were identified. Cerebellar atrophy, macrocephaly, and tall stature have been shown to coexist [51]. Other researchers have demonstrated the coexistence of neurodevelopmental disorders with benign short or tall stature, obesity, microcephaly, or macrocephaly in the case of de novo variants in the mutation of the FBXO11 F-Box protein [52].

Hydrocephalus untreated surgically affected the course of individual development in the studied group of children; unlike surgically treated hydrocephalus, it co-occurred with absolute macrocephaly and high stature. It has been shown that a reduction in the severity of neurological symptoms can be obtained in children and adolescents with normotensive hydrocephalus after surgery [53]. The first revolutionary techniques of surgical treatment, using valves to decompress hydrocephalus, date from 1949, and since 1990, endoscopic ventriculostomy has been introduced [54]. Tall stature requires the use of an appropriate valve structure to decompress hydrocephalus in adults [55]. Although long-term outcomes with children treated for hydrocephalus vary, treatment is ground-breaking for their survival and functioning. Currently, there is a possibility of prenatal treatment of myelomeningocele and hydrocephalus [56,57,58]. Fetal surgical repair of myelomeningocele has been associated with an improved early neurological outcome compared with postnatal operation [29,59]. The lack of neurosurgical treatment of hydrocephalus in the studied group of patients with NTD was associated with the coexistence of somatic development disorders.

However, there was no difference in the level of motor development assessed by the GMFCS scale in children and adolescents with neurodysfunction, operated and unoperated for hydrocephalus. It is the defect of severity of the central nervous system associated with hydrocephalus that can mainly determine the degree of psychomotor development disorders, not the neurosurgical operation itself undertaken to treat hydrocephalus [60]. Hydrocephalus that complicates extensive intraventricular bleeding in a preterm newborn despite receiving neurosurgical treatment can cause death [61]. For example, of 52 patients with myelomeningocele, 31 (59.6%) developed hydrocephalus that required a shunt. During the first year, seven patients (13.4%) required shunt revision. The cause of shunt revision was wound problem in one patient (1.9%), underdrainage in two patients (3.8%), infection in three patients (5.7%) and obstruction in another patient (1.9%) [62]. Ventricular shunting definitely improved the care of children with hydrocephalus, although shunt failures are extremely common and cause significant morbidity [63]. The results of intrapubic treatment of hydrocephalus are very promising [64]. It was considered that hydrocephalus untreated surgically affected the course of individual development in the studied group of children, in contrast with hydrocephalus surgically treated.

### 4.1. Clinical Implications

Untreated hydrocephalus surgically affected the course of individual development in the study group of children, in contrast with surgically treated hydrocephalus. In children and adolescents with neurodysfunction, the presence of treated and untreated hydrocephalus worsened the prognosis for motor development. The clinical implication is to encourage parents of patients with hydrocephalus, especially those with untreated hydrocephalus, to undergo follow-up brain imaging studies (MRI) and to attend follow-up visits to neurosurgeons to determine if there is a change in qualification for surgical treatment.

### 4.2. Limitations

The study was retrospective and data from medical records were scarce. From a clinical point of view, there are other factors that significantly affect developmental disorders. For example, hormonal balance should be taken into account in future prospective studies, clinical symptoms indicating cranial hypertension, results of brain imaging studies, types of neurosurgical procedures, and complications after hydrocephalus neurosurgical treatment. Other limitations of the investigation are also described in previously published articles that complement the results presented [9,10,11,15,16].

## 5. Conclusions

Surgically treated hydrocephalus occurred in patients with CP and NTDs, and untreated hydrocephalus was present only in patients with NTDs. Surgically treated hydrocephalus co-occurred more often with sasMMCandHCP than with CP. Surgically untreated hydrocephalus co-occurred more with sasMMC and hydrocephalus. Untreated and treated hydrocephalus occurred more frequently with NTDs than with CP. Surgically treated hydrocephalus did not co-occur with developmental disorders. However, surgically untreated hydrocephalus co-occurred with developmental disorders. Macrocephaly and surgically untreated hydrocephalus often occurred in the studied group of children. In this case, the traditionally used definition of macrocephaly better differentiated the relationship between macrocephaly and surgically untreated hydrocephalus. Absolute macrocephaly and surgically untreated hydrocephalus often co-occurred in the studied group of subjects. Tall stature more commonly co-occurred with surgically untreated hydrocephalus compared with all subjects. Establishing dependencies enabled the traditional definition to be applied.

Hydrocephalus, if not treated surgically, had a negative impact on the course of individual development in the studied group of children, in contrast to hydrocephalus treated with surgery.

## Figures and Tables

**Table 1 ijerph-19-05712-t001:** Co-occurrence of hydrocephalus treated surgically with disease entities and syndromes with neurodysfunction (Table 1A) and the occurrence of treated hydrocephalus in separate subgroups (Table 1B,C).

**A. Units and Syndromes Running with Neurodysfunction**	**Hydrocephalus Treated Surgically and Others (*p* < 0.001; Cp = 0.624)**				
	**Hydrocephalus Treated Surgically**		**Others**		**In Total**
	**N (%)**	**ASR**	**N (%)**	**ASR**	**N (%)**
Neurodegeneration with brain iron accumulation—mitochondrial protein associated neurodegeneration	0 (0.0)	−0.4	2 (0.7)	0.4	2 (0.6)
Pompe disease	0 (0.0)	−0.3	1 (0.3)	0.3	1 (0.3)
Long-chain 3-hydroxyacyl-coenzyme A dehydrogenase deficiency	0 (0.0)	−0.3	1 (0.3)	0.3	1 (0.3)
Smith–Lemli–Opitz syndrome	0 (0.0)	−0.3	1 (0.3)	0.3	1 (0.3)
Glucose transporter 1 deficiency	0 (0.0)	−0.3	1 (0.3)	0.3	1 (0.3)
Non-ketotichyperglycinemia	0 (0.0)	−0.3	1 (0.3)	0.3	1 (0.3)
SMEI, Dravet syndrome	0 (0.0)	−0.3	1 (0.3)	0.3	1 (0.3)
State after surgery of myelomeningocele and hydrocephalus	17 (65.4)	14.4	0 (0.0)	−14.4	17 (5.2)
State after surgery of myelomeningocele	0 (0.0)	−0.5	3 (1.0)	0.5	3 (0.9)
State after surgery of parietooocipitalmenigocele	0 (0.0)	−0.3	1 (0.3)	0.3	1 (0.3)
Arnold–Chiari malformation	0 (0.0)	−0.4	2 (0.7)	0.4	2 (0.6)
Isolated hydrocephalus	0 (0.0)	−0.3	1 (0.3)	0.3	1 (0.3)
Down syndrome	0 (0.0)	−1.0	11 (3.7)	1.0	11 (3.4)
Edward syndrome	0 (0.0)	−0.3	1 (0.3)	0.3	1 (0.3)
Phelan–McDermid syndrome	0 (0.0)	−0.4	2 (0.7)	0.4	2 (0.6)
Mowat–Wilson syndrome	0 (0.0)	−0.3	1 (0.3)	0.3	1 (0.3)
Angelman syndrome	0 (0.0)	−0.3	1 (0.3)	0.3	1 (0.3)
Di George syndrome	0 (0.0)	−0.3	1 (0.3)	0.3	1 (0.3)
46,XY,del(X)(q24)	0 (0.0)	−0.3	1 (0.3)	0.3	1 (0.3)
Cornelia de Lange syndrome	0 (0.0)	−0.3	1 (0.3)	0.3	1 (0.3)
Schwachman–Diamond syndrome	0 (0.0)	−0.3	1 (0.3)	0.3	1 (0.3)
Prader–Willi syndrome	0 (0.0)	−0.3	1 (0.3)	0.3	1 (0.3)
46 XX, add(2)(q25)	0 (0.0)	−0.3	1 (0.3)	0.3	1 (0.3)
46XX, del(12)(q24.21q24.23)	0 (0.0)	−0.3	1 (0.3)	0.3	1 (0.3)
Fetal alcohol syndrome	0 (0.0)	−0.3	1 (0.3)	0.3	1 (0.3)
Cerebral palsy	9 (34.6)	−4.6	230 (76.4)	4.6	239 (73.1)
Hereditary motor and sensory polyneuropathy	0 (0.0)	−0.8	8 (2.7)	0.8	8 (2.4)
Muscular dystrophy limb–girdle	0 (0.0)	−0.8	7 (2.3)	0.8	7 (2.1)
Becker’s muscular dystrophy	0 (0.0)	−0.5	3 (1.0)	0.5	3 (0.9)
Duchenne muscular dystrophy	0 (0.0)	−0.8	7 (2.3)	0.8	7 (2.1)
Thomsen disease	0 (0.0)	−0.3	1 (0.3)	0.3	1 (0.3)
Arthrogryposis multiplex congenital with neuropathy	0 (0.0)	−0.5	3 (1.0)	0.5	3 (0.9)
Congenital myopathy	0 (0.0)	−0.3	1 (0.3)	0.3	1 (0.3)
Spinal muscular atrophy	0 (0.0)	−0.4	2 (0.7)	0.4	2 (0.6)
In total	26 (100.0)		301 (100.0)		327 (100.0)
**B. Classification with respect to the etiopathogenesis, presence, and character of encephalopathy**	**Hydrocephalus treated surgically and others (*p* < 0.001; Cp = 0.549)**				
	**Hydrocephalus treated surgically**		**Others**		**In total**
	**N(%)**	**ASR**	**N(%)**	**ASR**	**N(%)**
Encephalopathy in metabolic disorder	0 (0.0)	−0.8	7 (2.3)	0.8	7 (2.1)
Epileptic encephalopathy	0 (0.0)	−0.3	1 (0.3)	0.3	1 (0.3)
Encephalopathy in neural tube defects	17 (65.4)	11.8	7 (2.3)	−11.8	24 (7.3)
Encephalopathy in genetic disorders	0 (0.0)	−1.5	23 (7.6)	1.5	23 (7.0)
Toxic encephalopathy	0 (0.0)	−0.3	1 (0.3)	0.3	1 (0.3)
Encephalopathy in cerebral palsy	9 (34.6)	−4.6	230 (76.4)	4.6	239 (73.1)
Neuromuscular disorders	0 (0.0)	−1.8	32 (10.6)	1.8	32 (9.8)
In total	26 (100.0)		301 (100.0)		327 (100.0)
**C. Classification with respect to** **presence and character of encephalopathy**	**Hydrocephalus treated surgically and others (*p* = 0.140; Cp = 0.109)**				
	**Hydrocephalus treated surgically**		**Others**		**In total**
	**N(%)**	**ASR**	**N (%)**	**ASR**	**N(%)**
Progressive encephalopathy	0 (0.0)	−0.8	8 (2.7)	0.8	8 (2.4)
Non progressive encephalopathy	26 (100.0)	2.0	261 (86.7)	−2.0	287 (87.8)
Neuromuscular disorders	0 (0.0)	−1.8	32 (10.6)	1.8	32 (9.8)
In total	26 (100.0)		301 (100.0)		327 (100.0)

Others—other patients without hydrocephalus treated surgically; N—numbers of patients; %—percent; *p*—probability value calculated by the chi-square test of independence; Cp—Pearson’s contingency coefficient C, Cp ≥ 0, values distant from 0 reflect a relationship, values approaching 1 correspond to a near-perfect association; ASR—adjusted standardized residuals, values > 1.96 reflect a higher number, and those below < −1.96 correspond to a lower number than a random distribution.

**Table 2 ijerph-19-05712-t002:** Co-occurrence of hydrocephalus untreated surgically with disease entities and syndromes with neurodysfunction (Table 2A) and occurrence of hydrocephalus untreated surgically in separate subgroups (Table 2B,C).

**A. Units and Syndromes Running with Neurodysfunction**	**Hydrocephalus Untreated Surgically and Others (*p* < 0.001; Cp = 0.661)**				
	**Hydrocephalus Untreated Surgically**		**Others**		**In Total**
	**N (%)**	**ASR**	**N (%)**	**ASR**	**N (%)**
Neurodegeneration with brain iron accumulation—mitochondrial protein associated neurodegeneration	0 (0.0)	−0.1	2 (0.6)	0.1	2 (0.6)
Pompe disease	0 (0.0)	−0.1	1 (0.3)	0.1	1 (0.3)
Long-chain 3-hydroxyacyl-coenzyme A dehydrogenase deficiency	0 (0.0)	−0.1	1 (0.3)	0.1	1 (0.3)
Smith–Lemli–Opitz syndrome	0 (0.0)	−0.1	1 (0.3)	0.1	1 (0.3)
Glucose transporter 1 deficiency	0 (0.0)	−0.1	1 (0.3)	0.1	1 (0.3)
Non-ketotichyperglycinemia	0 (0.0)	−0.1	1 (0.3)	0.1	1 (0.3)
SMEI, Dravet syndrome	0 (0.0)	−0.1	1 (0.3)	0.1	1 (0.3)
State after surgery of myelomeningocele and hydrocephalus	0 (0.0)	−0.4	17 (5.2)	0.4	17 (5.2)
State after surgery of myelomeningocele	2 (66.7)	12.0	1 (0.3)	−12.0	3 (0.9)
State after surgery of parietooocipitalmenigocele	0 (0.0)	−0.1	1 (0.3)	0.1	1 (0.3)
Arnold–Chiari malformation	0 (0.0)	−0.1	2 (0.6)	0.1	2 (0.6)
Isolated hydrocephalus	1 (33.3)	10.4	0 (0.0)	−10.4	1 (0.3)
Down syndrome	0 (0.0)	−0.3	11 (3.4)	0.3	11 (3.4)
Edward syndrome	0 (0.0)	−0.1	1 (0.3)	0.1	1 (0.3)
Phelan–McDermid syndrome	0 (0.0)	−0.1	2 (0.6)	0.1	2 (0.6)
Mowat–Wilson syndrome	0 (0.0)	−0.1	1 (0.3)	0.1	1 (0.3)
Angelman syndrome	0 (0.0)	−0.1	1 (0.3)	0.1	1 (0.3)
Di George syndrome	0 (0.0)	−0.1	1 (0.3)	0.1	1 (0.3)
46,XY,del(X)(q24)	0 (0.0)	−0.1	1 (0.3)	0.1	1 (0.3)
Cornelia de Lange syndrome	0 (0.0)	−0.1	1 (0.3)	0.1	1 (0.3)
Schwachman–Diamond syndrome	0 (0.0)	−0.1	1 (0.3)	0.1	1 (0.3)
Prader–Willi syndrome	0 (0.0)	−0.1	1 (0.3)	0.1	1 (0.3)
46 XX, add(2)(q25)	0 (0.0)	−0.1	1 (0.3)	0.1	1 (0.3)
46XX, del (12) (q24.21q24.23)	0 (0.0)	−0.1	1 (0.3)	0.1	1 (0.3)
Fetal alcohol syndrome	0 (0.0)	−0.1	1 (0.3)	0.1	1 (0.3)
Cerebral palsy	0 (0.0)	−2.9	239 (73.8)	2.9	239 (73.1)
Hereditary motor and sensory polyneuropathy	0 (0.0)	−0.3	8 (2.5)	0.3	8 (2.4)
Muscular dystrophy limb–girdle	0 (0.0)	−0.3	7 (2.2)	0.3	7 (2.1)
Becker muscular dystrophy	0 (0.0)	−0.2	3 (1.0)	0.2	3 (0.9)
Duchenne muscular dystrophy	0 (0.0)	−0.3	7 (2.2)	0.3	7 (2.1)
Thomsen disease	0 (0.0)	−0.1	1 (0.3)	0.1	1 (0.3)
Arthrogryposis multiplex congenital with neuropathy	0 (0.0)	−0.2	3 (1.0)	0.2	3 (0.9)
Congenital myopathy	0 (0.0)	−0.1	1 (0.3)	0.1	1 (0.3)
Spinal muscular atrophy	0 (0.0)	−0.1	2 (0.6)	0.1	2 (0.6)
In total	3 (100.0)		324 (100.0)		327 (100.0)
**B. Classification with respect to the etiopathogenesis, presence, and character of encephalopathy**	**Hydrocephalus untreated surgically and others (*p* < 0.001; Cp = 0.324)**				
	**Hydrocephalus untreated surgically**		**Others**		**In total**
	**N (%)**	**ASR**	**N (%)**	**ASR**	**N (%)**
Encephalopathy in metabolic disorder	0 (0.0)	−0.3	7 (2.2)	0.3	7 (2.1)
Epileptic encephalopathy	0 (0.0)	−0.1	1 (0.3)	0.1	1 (0.3)
Encephalopathy in neural tube defects	3 (100.0)	6.2	21 (6.5)	−6.2	24 (7.3)
Encephalopathy in genetic disorders	0 (0.0)	−0.5	23 (7.1)	0.5	23 (7.0)
Toxic encephalopathy	0 (0.0)	−0.1	1 (0.3)	0.1	1 (0.3)
Encephalopathy in cerebral palsy	0 (0.0)	−2.9	239 (73.8)	2.9	239 (73.1)
Neuromuscular disorders	0 (0.0)	−0.6	32 (10.6)	0.6	32 (9.8)
In total	3 (100.0)		324 (100.0)		327 (100.0)
**C. Classification with respect to** **presence and character of encephalopathy**	**Hydrocephalus untreated surgically and others (*p* = 0.810; Cp = 0.036)**				
	**Hydrocephalus untreated surgically**		**Others**		**In Total**
	**N (%)**	**ASR**	**N (%)**	**ASR**	**N (%)**
Progressive encephalopathy	0 (0.0)	−0.3	8 (2.5)	0.3	8 (2.4)
Non progressive encephalopathy	3 (100.0)	0.6	284 (87.7)	−0.6	287 (87.8)
Neuromuscular disorders	0 (0.0)	−0.6	32 (10.6)	0.6	32 (9.8)
In total	3 (100.0)		324 (100.0)		327 (100.0)

Others—all other patients without hydrocephalus untreated surgically; N—number of patients; %—percent; *p*—probability value calculated by the chi-square test of independence; Cp—Pearson’s contingency coefficient C, Cp ≥ 0, values distant from 0 reflect a relationship, values close to 1 correspond to a near-perfect association; ASR—adjusted standardized residuals, values > 1.96 reflect a higher number, and those below < −1.96 correspond to a lower number than a random distribution.

**Table 3 ijerph-19-05712-t003:** Co-occurrence of hydrocephalus treated surgically with developmental disorders.

**A. Head size—Dysmorphology Classification (hc)**	**Hydrocephalus Treated Surgically and Others (*p* = 0.121; Cp = 0.113)**				
	**Hydrocephalus Treated Surgically**		**Others**		**In total**
	**N (%)**	**ASR**	**N (%)**	**ASR**	**N (%)**
Normal head size	20 (76.9)	−0.9	253 (84.1)	0.9	273 (83.5)
Microcephaly	3 (11.5)	−0.2	38 (12.6)	0.2	41 (12.5)
Macrocephaly	3 (11.5)	2.1	10 (3.3)	−2.1	13 (4.0)
In total	26 (100.0)		301 (100.0)		327 (100.0)
**B. Head size—traditional classification (hc)**	**Hydrocephalus treated surgically and others (*p* = 0.448; Cp = 0.070)**				
	**Hydrocephalus treated surgically**		**Others**		**In total**
	**N (%)**	**ASR**	**N (%)**	**ASR**	**N (%)**
Normal head size	18 (69.2)	0.1	206 (68.4)	−0.1	224 (68.5)
Microcephaly	4 (15.4)	−0.9	68 (22.6)	0.9	72 (22.0)
Macrocephaly	4 (15.4)	1.1	27 (9.0)	−1.1	31 (9.5)
In total	26 (100.0)		301 (100.0)		327 (100.0)
**C. The abnormal head size—dysmorphology classification (hc&hc/h)**	**Hydrocephalus treated surgically and others (*p* = 0.051; Cp = 0.355)**				
	**Hydrocephalus treated surgically**		**Others**		**In total**
	**N (%)**	**ASR**	**N (%)**	**ASR**	**N (%)**
Relative microcephaly	2 (33.3)	−1.8	34 (70.8)	1.8	36 (66.7)
Absolute microcephaly	1 (16.7)	0.7	4 (8.3)	−0.7	5 (9.3)
Relative macrocephaly	0 (0.0)	−0.8	5 (10.4)	0.8	5 (9.3)
Absolute macrocephaly	3 (50.0)	2.6	5 (10.4)	−2.6	8 (14.8)
In total	6 (100.0)		48 (100.0)		54 (100.0)
**D. The abnormal head size—traditional classification (hc&hc/h)**	**Hydrocephalus treated surgically and others (*p* = 0.052; Cp = 0.264)**				
	**Hydrocephalus treated surgically**		**Others**		**In total**
	**N (%)**	**ASR**	**N (%)**	**ASR**	**N (%)**
Relative microcephaly	3 (37.5)	−1.2	57 (60.0)	1.2	60 (58.3)
Absolute microcephaly	1 (12.5)	0.1	11 (11.6)	−0.1	12 (11.7)
Relative macrocephaly	0 (0.0)	−1.2	14 (16.7)	1.2	14 (13.6)
Absolute macrocephaly	4 (50.0)	2.7	13 (13.7)	−2.7	17 (16.5)
In total	8 (100.0)		95 (100.0)		103 (100.0)
**E. Height—proposed classification (h)**	**Hydrocephalus treated surgically and others (*p* = 0.084; Cp = 0.122)**				
	**Hydrocephalus treated surgically**		**Others**		**In total**
	**N (%)**	**ASR**	**N (%)**	**ASR**	**N (%)**
Normal body height	14 (53.8)	−2.2	222 (73.8)	2.2	236 (72.2)
Short stature	12 (46.2)	2.2	78 (25.9)	−2.2	90 (27.5)
Tall stature	0 (0.0)	−0.3	1 (0.3)	0.3	1 (0.3)
In total	26 (100.0)		301 (100.0)		327 (100.0)
**F. Height—traditional classification (h)**	**Hydrocephalus treated surgically and others (*p* = 0.053; Cp = 0.133)**				
	**Hydrocephalus treated surgically**		**Others**		**In total**
	**N (%)**	**ASR**	**N (%)**	**ASR**	**N (%)**
Normal body height	12 (46.2)	−2.4	208 (69.1)	2.4	220 (67.3)
Short stature	13 (50.0)	2.4	84 (27.9)	−2.4	97 (29.7)
Tall stature	1 (3.8)	0.2	9 (3.0)	−0.2	10 (3.1)
In total	26 (100.0)		301 (100.0)		327 (100.0)
**G. Three categories of nutritional status**	**Hydrocephalus treated surgically and others (*p* = 0.131; Cp = 0.111)**				
	**Hydrocephalus treated surgically**		**Others**		**In total**
	**N (%)**	**ASR**	**N (%)**	**ASR**	**N (%)**
Normal state of nutrition	9 (34.6)	−0.8	130 (43.2)	0.8	139 (42.5)
Body mass deficiency in relation to height	8 (30.8)	−0.8	116 (38.5)	0.8	124 (37.9)
Excess body weight in relation to height	9 (34.6)	2.0	55 (18.3)	−2.0	64 (19.6)
In total	26 (100.0)		301 (100.0)		327 (100.0)
**H. Five categories of nutritional status**	**Hydrocephalus treated surgically and others (*p* = 0.114; Cp = 0.149)**				
	**Hydrocephalus treated surgically**		**Others**		**In total**
	**N (%)**	**ASR**	**N (%)**	**ASR**	**N (%)**
Normal state of nutrition	9 (34.6)	−0.8	130 (43.2)	0.8	139 (42.5)
Underweight	2 (7.7)	−1.6	63 (20.9)	1.6	65 (19.9)
Malnutrition	6 (23.1)	0.7	53 (17.6)	−0.7	59 (18.0)
Overweight	5 (19.2)	2.1	22 (7.3)	−2.1	27 (8.3)
Obesity	4 (15.4)	0.7	33 (11.0)	−0.7	37 (11.3)
In total	26 (100.0)		301 (100.0)		327 (100.0)

Others—all other patients without hydrocephalus treated surgically; N—numbers of patients; %—percent; *p*—probability value calculated by the chi-square test of independence; Cp—Pearson contingency coefficient C, Cp ≥ 0, values distant from 0 reflect a relationship, values approaching 1 correspond to a near-perfect association; ASR—adjusted standardized residuals, values > 1.96 reflect a higher number, and those below < −1.96 correspond to a lower number than a random distribution.

**Table 4 ijerph-19-05712-t004:** Co-occurrence of hydrocephalus untreated surgically with developmental disorders.

**A. Head Size—Dysmorphology Classification (hc)**	**Hydrocephalus Untreated Surgically and Others (*p* = 0.030; Cp = 0.145)**				
	**Hydrocephalus Untreated Surgically**		**Others**		**In Total**
	**N (%)**	**ASR**	**N (%)**	**ASR**	**N (%)**
Normal head size	2 (66.7)	−0.8	271 (83.6)	0.8	273 (83.5)
Microcephaly	0 (0.0)	−0.7	41 (12.7)	0.7	41 (12.5)
Macrocephaly	1 (33.3)	2.6	12 (3.7)	−2.6	13 (4.0)
In total	3 (100.0)		324 (100.0)		327 (100.0)
**B. Head size—traditional classification (hc)**	**Hydrocephalus untreated surgically and others (*p* < 0.001; Cp = 0.285)**				
	**Hydrocephalus untreated surgically**		**Others**		**In total**
	**N (%)**	**ASR**	**N (%)**	**ASR**	**N (%)**
Normal head size	0 (0.0)	−2.6	224 (69.1)	2.6	224 (68.5)
Microcephaly	0 (0.0)	−0.9	72 (22.2)	0.9	72 (22.0)
Macrocephaly	3 (100.0)	5.4	28 (8.6)	−5.4	31 (9.5)
In total	3 (100.0)		324 (100.0)		327 (100.0)
**C. The abnormal head size—dysmorphology classification (hc&hc/h)**	**Hydrocephalus untreated surgically and others (*p* = 0.119; Cp = 0.313)**				
	**Hydrocephalus untreated surgically**		**Others**		**In total**
	**N (%)**	**ASR**	**N (%)**	**ASR**	**N (%)**
Relative microcephaly	0 (0.0)	−1.4	36 (67.9)	1.4	36 (66.7)
Absolute microcephaly	0 (0.0)	−0.3	5 (9.4)	0.3	5 (9.3)
Relative macrocephaly	0 (0.0)	−0.3	5 (9.4)	0.3	5 (9.3)
Absolute macrocephaly	1 (100.0)	2.4	7 (13.2)	−2.4	8 (14.8)
In total	1 (100.0)		53 (100.0)		54 (100.0)
**D. The abnormal head size—traditional classification (hc&hc/h)**	**Hydrocephalus untreated surgically and others (*p* = 0.001; Cp = 0.363)**				
	**Hydrocephalus untreated surgically**		**Others**		**In total**
	**N (%)**	**ASR**	**N (%)**	**ASR**	**N (%)**
Relative microcephaly	0 (0.0)	−2.1	60 (60.0)	2.1	60 (58.3)
Absolute microcephaly	0 (0.0)	−0.6	12 (12.0)	0.6	12 (11.7)
Relative macrocephaly	0 (0.0)	−0.7	14 (14.0)	0.7	14 (13.6)
Absolute macrocephaly	3 (100.0)	4.0	14 (14.0)	−4.0	17 (16.5)
In total	3 (100.0)		100 (100.0)		103 (100.0)
**E. Height—proposed classification (h)**	**Hydrocephalus untreated surgically and others (*p* = 0.558; Cp = 0.060)**				
	**Hydrocephalus untreated surgically**		**Others**		**In total**
	**N (%)**	**ASR**	**N (%)**	**ASR**	**N (%)**
Normal body height	233 (71.9)	−1.1	3 (100.0)	1.1	236 (72.2)
Short stature	90 (27.8)	1.1	0 (0.0)	−1.1	90 (27.5)
Tall stature	1 (0.3)	0.1	0 (0.0)	−0.1	1 (0.3)
In total	3 (100.0)		324 (100.0)		327 (100.0)
**F. Height—traditional classification (h)**	**Hydrocephalus untreated surgically and others (*p* = 0.007; Cp = 0.172)**				
	**Hydrocephalus untreated surgically**		**Others**		**In total**
	**N (%)**	**ASR**	**N (%)**	**ASR**	**N (%)**
Normal body height	218 (67.3)	0.0	2 (66.7)	0.0	220 (67.3)
Short stature	97 (29.9)	1.1	0 (0.0)	−1.1	97 (29.7)
Tall stature	9 (2.8)	−3.1	1 (33.3)	3.1	10 (3.1)
In total	3 (100.0)		324 (100.0)		327 (100.0)
**G. Three categories of nutritional status**	**Hydrocephalus untreated surgically and others (*p* = 0.098; Cp = 0.118)**				
	**Hydrocephalus untreated surgically**		**Others**		**In total**
	**N (%)**	**ASR**	**N (%)**	**ASR**	**N (%)**
Normal state of nutrition	1 (33.3)	−0.3	138 (42.6)	0.3	139 (42.5)
Body mass deficiency in relation to height	0 (0.0)	−1.4	124 (38.3)	1.4	124 (37.9)
Excess body weight in relation to height	2 (66.7)	2.1	62 (19.1)	−2.1	64 (19.6)
In total	3 (100.0)		324 (100.0)		327 (100.0)
**H. Five categories of nutritional status**	**Hydrocephalus untreated surgically and others (*p* = 0.307; Cp = 0.120)**				
	**Hydrocephalus untreated surgically**		**Others**		**In total**
	**N (%)**	**ASR**	**N (%)**	**ASR**	**N (%)**
Normal state of nutrition	1 (33.3)	−0.3	138 (42.6)	0.3	139 (42.5)
Underweight	0 (0.0)	−0.9	65 (20.1)	0.9	65 (19.9)
Malnutrition	0 (0.0)	−0.8	59 (18.2)	0.8	59 (18.0)
Overweight	1 (33.3)	1.6	26 (8.0)	−1.6	27 (8.3)
Obesity	1 (33.3)	1.2	36 (11.1)	−1.2	37 (11.3)
In total	3 (100.0)		324 (100.0)		327 (100.0)

Others—all other patients without hydrocephalus untreated surgically; N—numbers of patients; %—percent; *p*—probability value calculated by the chi-square test of independence; Cp—Pearson’s contingency coefficient C, Cp ≥ 0, values distant from 0 reflect a relationship, values approaching 1 correspond to a near-perfect association; ASR—adjusted standardized residuals, values > 1.96 reflect a higher number, and those below < −1.96 correspond to a lower number than a random distribution.

**Table 5 ijerph-19-05712-t005:** GMFCS levels, GMFCS score and hydrocephalus treated and untreated surgically.

**Levels of the Gross Motor Function Classification System (*p* = 0.133, Cp = 0.150)**											
**A.**	**I**		**II**		**III**		**IV**		**V**		
	N (%)	ASR	N(%)	ASR	N (%)	ASR	N (%)	ASR	N (%)	ASR	In Total
Hydrocephalus treated and untreated surgically	3 (10.3)	−1.8	10 (34.5)	−0.7	6 (20.7)	1.7	6 (20.7)	1.3	4 (13.8)	0.4	29 (100.0)
Without hydrocephalus	74 (24.8)	1.8	124 (41.6)	0.7	30 (10.1)	−1.7	36 (12.1)	−1.3	34 (11.4)	−0.4	298 (100.0)
In total	77 (23.5)		134 (41.0)		36 (11.0)		42 (12.8)		38 (11.6)		327 (100.0)
**Levels of the Gross Motor Function Classification System (*p* = 0.335, Cp = 0.370)**											
**B.**	**I**		**II**		**III**		**IV**		**V**		
	N (%)	ASR	N (%)	ASR	N (N%)	ASR	N (%)	ASR	N (%)	ASR	In Total
**Hydrocephalus treated surgically**	2 (7.7)	−1.4	8 (30.8)	−1.2	6 (23.1)	0.9	6 (23.1)	0.9	4 (15.4)	0.7	26 (100.0)
**Hydrocephalus untreated surgically**	1 (33.3)	1.4	2 (66.7)	1.2	0 (0.0)	−0.9	0 (0.0)	−0.9	0 (0.0)	−0.7	3 (100.0)
In total	3 (10.3)		10 (34.5)		6 (20.7)		6 (20.7)		4 (13.8)		29 (100.0)
**Levels of the Gross Motor Function Classification System (*p* = 0.060, Cp = 0.130)**											
**C.**	**A (I + II)**				**B (III)**		**C (IV + V)**				
	N (%)		ASR		N (%)	ASR	N (%)		ASR		In Total
**Hydrocephalus treated and untreated surgically**	13 (44.8)		−2.3		6 (20.7)	1.7	10 (34.5)		1.3		29 (100.0)
**Without hydrocephalus**	197 (66.1)		2.3		31 (10.4)	−1.7	70 (23.5)		−1.3		298 (100.0)
In total	210 (64.2)				37 (11.3)		80 (24.5)				327 (100.0)
**Levels of the Gross Motor Function Classification System (*p* = 0.128, Cp = 0.350)**											
**D.**	**A (I + II)**				**B (III)**		**C (IV + V)**				
	N (%)		ASR		N (%)	ASR	N (%)		ASR		In Total
**Hydrocephalus treated surgically**	10 (38.5)		−2		6 (23.1)	0.9	10 (38.5)		1.3		26 (100.0)
**Hydrocephalus untreated surgically**	3 (100.0)		2		0 (0.0)	−0.9	0 (0.0)		−1.3		3 (100.0)
In total	13 (44.8)				6 (20.7)		10 (34.5)				29 (100.0)
**E.**	**Subgroups**										**Mann–Whitney test**
Gross Motor Function Classification System **I-V score**	**Without hydrocephalus** **(N = 298)**					**Hydrocephalus treated & untreated surgically (N = 29)**					
Mean ± *s*	2.44 ± 1.29					2.93 ± 1.25					*p* = 0.027
Median	2					3					
Guartiles	2–3					2–4					
**F.**	**Subgroups**										**Mann–Whitney test**
Gross Motor Function Classification System **I-V score**	**Hydrocephalus treated surgically** **(N = 26)**					**Hydrocephalus untreated surgically (N = 3)**					
Mean ± *s*	3.08 ± 1.23					1.67 ± 0.58					*p* = 0.059
Median	3					2					
Guartiles	2–4					1.5–2					
**G.**	**Subgroups**										**Mann–Whitney test**
Gross Motor Function Classification System **A-C score**	**Without hydrocephalus** **(N = 298)**					**Hydrocephalus treated & untreated surgically (N = 29)**					
Mean ± *s*	1.57 ± 0.85					1.9 ± 0.9					*p* = 0.036
Median	1					2					
Guartiles	1–2					1–3					
**H.**	**Subgroups**										**Mann–Whitney test**
Gross Motor Function Classification System **A-C score**	**Hydrocephalus treated surgically** **(N = 26)**					**Hydrocephalus untreated surgically (N = 3)**					
Mean ± *s*	2 ± 0.89					1 ± 0					*p* = 0.070
Median	2					1					
Guartiles	1–3					1–1					

N—number of patients; %—percent; *p*—probability value calculated by the chi-square test of independence; Cp—Pearson’s contingency coefficient C, Cp ≥ 0, values distant from 0 reflect a relationship, values approaching 1 correspond to a near-perfect association; ASR—adjusted standardized residuals, values > 1.96 reflect a higher number, and those below < −1.96 correspond to a lower number than a random distribution.

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
