# Peer review of "Somatic Development Disorders in Children and Adolescents Affected by Syndromes and Diseases Associated with Neurodysfunction and Hydrocephalus Treated/Untreated Surgically"

_ijerph, 2022, doi:10.3390/ijerph19095712_

Round 1

Reviewer 1 Report

The study is a continued version of the prior clinical observation related with children with various diseases. Authors have investigated hydrocephalus of children with or without treatment and compared the consequences thereof. Table 1 seems to be the most important data set and I have a few questions:

Major questions to Table 1:

  1. Lack - does this mean patients (are) "lacking" surgical treatment?
  2. Present - does this mean patients (are) treated with surgical shunt etc?

Provided that the above understanding is correct, 2 key messages are found:

3. If infants left untreated ==> 76% will manifest with cerebral palsy

4. If infants treated w/ surgery ==> 34 % will have cerebral palsy while 65% manifest with hydrocephalus (due to myelomeningocele)

If this is correct, please revise the description on this finding in your abstract as well as in the conclusion. 

Author Response

Response to Reviewer I

First of all, we would like to thank the Reviewer for the commentary on the content of the article as well as tips and suggestions for introducing changes and corrections. These changes will undoubtedly contribute to increasing the scientific value of the article. Our response refers to the individual guidelines in the order in which they appear in the Reviewer's comment. All changes and corrections are marked in red colour. As it can be seen, the authors made extensive changes to the text, especially in the 'Results' section. We hope that this will make the perception of the article easier for the potential reader.

Major questions to Table 1:

  1. Lack - does this mean patients (are) "lacking" surgical treatment?

Response: The primary diagnosis is provided in the first column of Table 1A. ‘None’ - means no additional diagnosis such as, for example,  hydrocephalus treated surgically. These are all other patients in the study group who did not have additional diagnosis ‘hydrocephalus treated surgically’, and had the diagnosis from the first column of Table 1A.

  1. Present - does this mean patients (are) treated with surgical shunt etc?

Response: The primary diagnosis is provided in the first column of Table 1A. ‘Present’ - means the presence of additional diagnosis such as ‘hydrocephalus treated surgically’. These are therefore patients who had a primary diagnosis as listed in column 1 of Table 1A and an additional diagnosis of hydrocephalus treated surgically.

In connection with the reviewer’s I remarks, not only the Table I was redrafted, but also the others, the order of the columns was changed, the second column was replaced by the third column, the third column was replaced by the second column, the text in the tables was corrected to make it more understandable. The terms ‘none’ and ‘present’ have been removed. Descriptive terms are used, for example ‘hydrocephalus treated surgically’ or shortened forms; for example ‘others’ (as explained under the table ‘all other patients without hydrocephalus treated surgically’).

  1. If infants left untreated ==> 76% will manifest with cerebral palsy
  2. If infants treated w/ surgery ==> 34 % will have cerebral palsy while 65% manifest with hydrocephalus (due to myelomeningocele).

If this is correct, please revise the description on this finding in your abstract as well as in the conclusion. 

Response: The authors are not able to establish a strict, unequivocal cause-and-effect relationship on the basis of the collected data, but taking into account the above very important guidelines, the following observations were made:

Nine (3.8%) patients out of 239 with cerebral palsy had an additional diagnosis of hydrocephalus treated surgically (Table 1A, 1B). Out of 24 patients with neural tube defects, 17 (70.8%) also had the diagnosis of hydrocephalus treated surgically (Table 1B), and 3 (12.5%) out of 24 patients with neural tube defects - hydrocephalus untreated surgically (Table 2B). In the neural tube defects subgroup, the total of patients with hydrocephalus treated and untreated surgically was 83.3% (20 patients out of 24). Among patients with additional diagnosis ‘hydrocephalus treated surgically’ (N=26) as primary diagnosis, 34.6% had cerebral palsy (N=9) and 65.4% had neural tube defects (patients operated for both myelomeningocele and hydrocephalus, N=17) (Table 1A, 1B). Hydrocephalus untreated surgically (N=3) only occurred in the subgroup of patients with neural tube defects (patients operated only for myelomeningocele: N=2, 66.7%, isolated hydrocephalus: N=1, 33.3%) (Table 2A, 2B).

The above content is included in the ‘Results’ section.

The following information has been added to the indicated sections of the article: 9 (3.8%) of 239 patients with cerebral palsy had hydrocephalus treated surgically, 17 (70.8%) of 24 patients with neural tube defects also had hydrocephalus treated surgically, and 3 (12.5%) of 24 patients with neural tube defects - hydrocephalus untreated surgically (‘Abstract’ section). Surgically treated hydrocephalus occurred in patients with cerebral palsy and neural tube defects, and untreated hydrocephalus was present only in patients with neural tube defects.

Once again, we would like to thank very much for all the suggestions for improving this article.

Reviewer 2 Report

The reviewed paper was conducted to evaluate co-occurrence of hydrocephalus treated/untreated surgically and congenital nervous system disorders or neurological syndromes with symptoms visible since infancy, and with somatic development disorders, based on significant data obtained during admission to a neurological rehabilitation unit for children and adolescents. The authors have performed a retrospective analysis of 327 children and adolescents (age range 14 – 18 years in abstract, 4 – 18 years in the main text – please correct in the Abstract) all presenting with congenital disorders of the nervous system and/or neurological syndromes associated with at least one neurodysfunction existing from early childhood. Criteria to enable identification of individuals with somatic development disorders were adopted. Treated/untreated hydrocephalus was found in the study group at the rates of 8% and 0.9%, respectively. As expected hydrocephalus is a more frequent comorbidity in subjects with neural tube defects compared to those with cerebral palsy. Also subjects with untreated hydrocephalus more commonly present with macrocephaly, including absolute macrocephaly, and with tall stature. The authors conclude that untreated hydrocephalus changes the course of individual development (the data from the Abstract support this conclusion only for head size and tall stature, maybe more information should be added i the abstract) in the studied group of children, in contrast to surgically treated hydrocephalus.

Although the author is not a native speaker of the English language, the paper deserves language correction by a native speaker. However the topic – the association of hydrocephalus with various types of neurological disorders in infanty/adolescence is definitively worth studying from the neurosurgical point of view. To simplify the problem – in case of unfavourable course of the disease the malfunction of the implanted shunt system is usually blamed leading to unnecessary and potentially harmful shunt revisions. However the main aim of the study is to verify the hypothesis that untreated hydrocephalus affects the defined parameters of the individual development of the patient. The inclusion/ exclusion criteria are adequately and comprehensively defined (with one exception – how was the disctinction between active hydrocephalus requiring treatment and simple ventriculomegaly not requiring surgery). The ethical problems are adequately resolved. Regarding the Results section: The system of abbreviation is rather complex, all abbreviations are adequately explained, but the reader sometimes becomes lost in the complexity of abbreviations. Similarly the Tables 1- 5 provides rather complicated overview of the results. Therefore I can only suggest to the authors to try to simplify their Results section to make it more eligible for potential readers. From neurosurgical point of view the percentages of shunt complications should be also added. However there is one clear take home message – early treatment of hydrocephalus in well indicated paediatric patients with various neurological disorders – and the authors have provided (although rather overcomplicated) overview of indication criteria and the expected clinical outcome. The authors are well aware of the limitations of their study (e.g. hormonal values). The authors conclude that treated and untreated hydrocephalus occurs more frequently with NTDs than with CP. Surgically treated hydrocephalus does not co-occur with developmental disorders. However, surgically untreated hydrocephalus co-occurs with developmental disorders. Not surprisingly macrocephaly and surgically untreated hydrocephalus often occur in the studied group. The most important message is that hydrocephalus  untreated surgically affects the course of individual development in the studied group of children, in contrast to surgically treated hydrocephalus. Therefore from the simple neurosurgical point of view the paper can be recommended for publication but the authors should address two points – language correction (the level of the language is acceptable and the Reviewer is not a native speaker of English language) and try to simplify the way they present their results, if possible.   

Author Response

Response to Reviewer II

First of all, we would like to thank the Reviewer for his extensive commentary on the content of the article as well as tips and suggestions for introducing changes and corrections. These changes will undoubtedly contribute to increasing the scientific value of the article. Our response refers to the individual guidelines in the order in which they appear in the Reviewer's comment. All changes and corrections are marked in red colour. As it can be seen, the authors made extensive changes to the text, especially in the 'Results' section. We hope that this will make the perception of the article easier for the potential reader.

  1. The error in the Abstract, mentioned by the Reviewer, has been corrected, i.e. the age has been changed from 14 to 4 according to reality.
  2. The following information has been added to the Abstract section:
  3. In 9 (3.8%) of 239 patients with cerebral palsy hydrocephalus was treated surgically, also 17 (70.8%) of 24 patients with neural tube defects had hydrocephalus treated surgically, and in 3 (12.5%) out of 24 patients with neural tube defects - hydrocephalus was untreated surgically (see ‘Results’ section).
  4. Excessive body mass co-occured more frequently with surgically untreated hydrocephalus, but the relationship was not statistically significant (p=0.098) (see ‘Results’ section).
  5. Surgically treated hydrocephalus occurred in patients with cerebral palsy and neural tube defects, and untreated hydrocephalus was present only in patients with neural tube defects.

  1. The article was submitted for linguistic proofreading to a person with extensive linguistic and research experience. In the event of insufficient linguistic proofreading, the team will use the publisher's linguistic proofreading. However, we are convinced that this is not necessary in the current form of the text

  1. The research team only conducted retrospective studies. In principle, patients admitted for rehabilitation in stationary conditions may not have symptoms of intracranial hypertension - they would not undergo the admission procedure. Patients have the described diagnoses: condition after surgery due to hydrocephalus or normotensive hydrocephalus. These diagnoses, like the others, is made by specialist doctors prior to admission to the ward. Such an approach (recording diagnoses) is necessary to define the rehabilitation program, determine contraindications to rehabilitation and to immediately react in the event of symptoms of intracranial hypertension (neurosurgical consultation, diagnostics, qualification for surgery, discharge from the department to the indicated unit). Therefore, we were able to analyze whether treated / untreated hydrocephalus differentiates the development of neurosurgical children. This information was entered in the ’Materials and methods’ section (‘Participants’). Data other than the diagnoses were not analyzed. This fact was noted in the ‘Limitations’ section.
  2. The ‘Results’ section has been redrafted. Results were organized and simplified, according to the Reviewer’s suggestion. The results are presented in the following paragraphs:

3.1. Basic percentage analysis

3.2. Co-occurrence of hydrocephalus treated surgically with diseases and syndromes associated with neurodysfunction and occurrence of treated hydrocephalus in the separate subgroups

3.3. Co-occurrence of hydrocephalus untreated surgically with diseases and syndromes associated with neurodysfunction and occurrence of untreated hydrocephalus in the separate subgroups

3.4. Co-occurrence of hydrocephalus treated surgically with developmental disorders

3.5. Co-occurrence of hydrocephalus untreated surgically with developmental disorders

3.6. Gross Motor Function Classification System and hydrocephalus treated and untreated surgically

In section (3.1.-3.6.) There is a description of the results in line with the section title and without abbreviations related to the names of diseases / syndromes / subgroups, taking into account statistical significance (sections 3.2.-3.6.), The discussed results are presented in the table (sections 3.2.-3.6.), most abbreviations have been omitted in the tables (Table 1-5, sections 3.2.-3.6.).

  1. Researchers did not have access to information on complications after neurosurgical treatment due to hydrocephalus. This information was entered in the ‘Limitations’ section.

The problem of valve complications was discussed in the ‘Discussion’ section.

However, there was no difference in the level of motor development as assessed by the GMFCS scale in children and adolescents with neurodysfunction, operated and unoperated for hydrocephalus. It is the severity of the central nervous system defect associated with hydrocephalus that may mainly determine the degree of psychomotor development disorders, and not the neurosurgical operation itself undertaken to treat hydrocephalus [60]. Hydrocephalus complicating extensive intraventricular bleeding in a preterm newborn despite receiving neurosurgical treatment may lead to death [61]. For example, of 52 patients with myelomeningocele, 31 (59.6%) developed hydrocephalus requiring a shunt. During the first year 7 patients (13.4%) required of revision of shunt. The cause of shunt revision was wound problem in 1 patient (1.9%), underdrainage in 2 patients (3.8%), infection in 3 patients (5.7%), and obstruction in another 1 patient (1.9%) [62]. Ventricular shunting definitely improved the care of children with hydrocephalus although shunt malfunctions are extremely common and cause significant morbidity [63]. The results of the intrapubic treatment of hydrocephalus are very promising [64]. Consider that hydrocephalus untreated surgically affects the course of individual development in the studied group of children, in contrast to surgically treated hydrocephalus.

Due to the expansion of the 'Discussion' section, the literature items listed below have been added to the 'References' section.

  1. Yamasaki, M.; Nonaka, M.; Bamba, Y.; Teramoto, C.; Ban, C.; Pooh, R.K. Diagnosis, treatment, and long-term outcomes of fetal hydrocephalus. Semin. Fetal Neonatal. Med. 2012, 17, 330–335. https://doi.org/10.1016/j.siny.2012.07.004.
  2. Dvalishvili, A.; Khinikadze, M.; Gegia, G.; Orlov, M. Comparative analysis of neurosurgical aspects of neonatal intraventricular hemorrhage treatment. Georgian Med. News. 2021, 320, 41–46.
  3. Alatas, I.; Canaz, G.;, Kayran, N.A.; Kara, N.; Canaz, H. Shunt revision rates in myelomeningocele patients in the first year of life: a retrospective study of 52 patients. Childs Nerv Syst. 2018, 34, 919–923. https://doi.org/10.1016/j.siny.2012.07.004.
  4. Liptak, G.S.; Masiulis B.S.; McDonald, J.V. Ventricular shunt survival in children with neural tube defects. Acta Neurochir. (Wien). 1985, 74, 113–117. https://doi.org/10.1007/BF01418798.
  5. Cavalheiro, S.; da Costa, MDS.; Barbosa, M.M.; Dastoli, P.A.; Mendonça J.N., Cavalheiro, D.; Moron, A.F. Hydrocephalus in myelomeningocele. Childs Nerv. Syst. 2021, 37, 3407–3415. https://doi.org/10.1007/s00381-021-05333-2.

The introduction of amendments proposed by the Reviewer made the article appear to have a higher scientific value, and at the same time it is more accessible to a potential reader. Therefore, we would like to thank the Reviewer once again for his/her help.

Round 2

Reviewer 1 Report

Authors have addressed the concerns raised. I have no further objection or comment.